# Characterization of Cognitive, Language and Adaptive Profiles of Children and Adolescents with Malan Syndrome

**DOI:** 10.3390/jcm11144078

**Published:** 2022-07-14

**Authors:** Paolo Alfieri, Marina Macchiaiolo, Martina Collotta, Federica Alice Maria Montanaro, Cristina Caciolo, Francesca Cumbo, Paolo Galassi, Filippo Maria Panfili, Fabiana Cortellessa, Marcella Zollino, Maria Accadia, Marco Seri, Marco Tartaglia, Andrea Bartuli, Corrado Mammì, Stefano Vicari, Manuela Priolo

**Affiliations:** 1Child and Adolescent Neuropsychiatry Unit, Department of Neuroscience, Bambino Gesù Children’s Hospital, IRCCS, 00165 Rome, Italy; martina.collotta@opbg.net (M.C.); federica.montanaro@opbg.net (F.A.M.M.); cristina.caciolo@opbg.net (C.C.); francesca.cumbo@opbg.net (F.C.); paolo.galassi@opbg.net (P.G.); stefano.vicari@opbg.net (S.V.); 2Rare Diseases and Medical Genetics Unit, University-Hospital Pediatric Department (DPUO), Bambino Gesù Children’s Hospital, IRCCS, 00165 Rome, Italy; marina.macchiaiolo@opbg.net (M.M.); fabiana.cortellessa@opbg.net (F.C.); andrea.bartuli@opbg.net (A.B.); 3Academic Department of Pediatrics, Bambino Gesù Children’s Hospital, IRCCS, 00165 Rome, Italy; filippomaria.panfili@opbg.net; 4Genetica Medica, Fondazione Policlinico Universitario A. Gemelli, IRCCS, 00168 Rome, Italy; marcella.zollino@unicatt.it; 5Dipartimento Universitario Scienze della Vita e Sanità Pubblica, Sezione di Medicina Genomica, Università Cattolica del Sacro Cuore Facoltà di Medicina e Chirurgia, 00168 Roma, Italy; 6Medical Genetics Service, Hospital “Cardinale G. Panico”, 73039 Tricase, Italy; m.accadia@piafondazionepanico.it; 7Unit of Medical Genetics, Azienda Ospedaliero Universitaria di Bologna, IRCCS, 40126 Bologna, Italy; marco.seri@unibo.it; 8Genetics and Rare Diseases Research Division, Bambino Gesù Children’s Hospital, IRCCS, 00165 Rome, Italy; marco.tartaglia@opbg.net; 9Operative Unit of Medical Genetics Bianchi-Melacrino-Morelli Great Metropolitan Hospital, 89133 Reggio Calabria, Italy; corradomammi@tiscali.it (C.M.); prioloma@libero.it (M.P.); 10Department of Life Sciences and Public Health, Università Cattolica del Sacro Cuore, 00168 Rome, Italy

**Keywords:** Malan Syndrome, *NFIX* variants, adaptive behavior, cognition, sensory processing, intellectual disability

## Abstract

Malan Syndrome (MS) is an ultra-rare overgrowth genetic syndrome due to heterozygous variants or deletions in the Nuclear Factor I X *(NFIX*) gene. It is characterized by an unusual facial phenotype, generalized overgrowth, intellectual disability (ID) and behavioral problems. Even though limitations in cognitive and adaptive functioning have been previously described, systematic studies on MS cohorts are still lacking. Here, we aim to define the cognitive and adaptive behavior profile of MS children and adolescents, providing quantitative data from standardized evaluations. Subjects included in this study were evaluated from October 2020 to January 2022 and the study is based on a retrospective data archive: fifteen MS individuals were recruited and underwent evaluation with Wechsler Intelligence Scales, Leiter International Performance Scales and Griffith Mental Development Scales for cognitive profiles and with Vineland Adaptive Behavior Scales-II Edition (VABS-II) for adaptive functioning. Language skills and visuomotor integration abilities were assessed too. Comparisons and correlations between scales and subtests were performed. All the assessed MS individuals showed both low cognitive and adaptive functioning. One subject presented with mild ID, five had moderate ID and eight showed severe ID. One female toddler received a diagnosis of psychomotor delay. Linguistic skills were impaired in all individuals, with language comprehension relatively more preserved. Results revealed significant differences between VABS-II subdomains and a strong relationship between cognitive and adaptive functioning. All subjects exhibited mild to moderate ID and adaptive behavior lower than normal, with communication skills being the most affected. Regarding the daily living skills domain, personal and community subscale scores were dramatically lower than for the domestic subdomain, highlighting the importance of considering behavior within developmental and environmental contexts. Our cognitive and adaptive MS characterization provides a more accurate quantitative MS profiling, which is expected to help clinicians to better understand the complexity of this rare disorder.

## 1. Introduction

Malan Syndrome (MS) (MIM #614753) is an autosomal dominant ultra-rare overgrowth condition caused by haploinsufficiency of the Nuclear Factor I X gene (*NFIX*; MIM #164005) due to heterozygous loss-of-function (LoF) pathogenic variants almost exclusively clustering within exon 2 of *NFIX* with few exceptions or to chromosomal microdeletions involving the 19p13.2 region. Based on the number of known affected individuals, MS prevalence is estimated as 1/1,000,000 [1,2].

*NFIX* encodes a transcription factor belonging to the Nuclear Factor I (NFI) family, which is involved in signal transduction and transcriptional regulation during brain and musculoskeletal development [3,4,5]. No significant clinical differences have been observed among individuals with intragenic *NFIX* variants or *NFIX* gene deletions, except for a significantly higher frequency of epilepsy and EEG anomalies in MS individuals carrying *NFIX* microdeletions. This observation has been explained by the presence of a contiguous gene disorder [2].

MS is allelic to Marshall–Smith syndrome (MSS, MIM #602535), which is characterized by a typical facial gestalt, failure to thrive, short stature, dysostosis, progressive scoliosis, respiratory compromise and moderate to severe developmental delay [6,7,8].

In MSS, *NFIX* variants are clustered at the 3′-prime exons (frameshift variants or intragenic deletions mostly confined to exons 6 to 7) and escape nonsense-mediated mRNA decay (NMD), leading to abnormal proteins with an aberrant shared C-terminus tail and preserved DNA binding and dimerization domains, resulting in a possible dominant negative effect [1,2,8].

According to Priolo et al. [2], the main features of MS are:-Generalized overgrowth. MS individuals may present with prenatal overgrowth resulting as Large for Gestational Age (LGA) newborns with birth weight >2 standard deviations (SDS) in 14.6% of cases. More frequently, they may present with a post-natal overgrowth, typically in childhood and adolescence, with a height >2 SDS reported in 56% of patients. Final height during adulthood is less marked, falling within two SDS of the mean in two-thirds of individuals;-Macrocephaly. This sign is observed in more than 75% of individuals;-Facial gestalt. Long and narrow face with a triangular shape (84%), high and prominent forehead (96%), short nose with anteverted nares (48%), everted lower lip and small mouth (54%), prognathia/prominent chin (74.5%), and blue sclerae (28.7%) are the facial hallmarks of the condition. Other less frequent features are highly arched palate and dental crowding, sparse hair, loose and soft skin, and facial asymmetry;-Neurological features. Hypotonia is observed in 76% of individuals. Seizures and EEG anomalies are common and more frequently observed among the population with *NFIX* microdeletions. Central nervous system (CNS) anomalies are characterized by wide ventricles, corpus callosum hypoplasia, Chiari malformation and, less frequently, brain atrophy. Optic nerves hypoplasia has been reported, as well;-Muscolo-skeletal anomalies. MS individuals may have advanced bone age. Scoliosis, hyper-kyphosis or hyper-lordosis, pectus excavatum/carinatum, slender habitus, and long hands are extremely frequent;-Ophthalmological features. Strabismus and refractive disorders, such as myopia, hypermetropia, and astigmatism are quite common. Nystagmus has been reported as well;-Cardiovascular diseases. Four patients with aortic root dilatation and one patient with pulmonary artery dilatation were reported [2,9,10];-Most individuals diagnosed with MS carry a de novo genetic *NFIX* pathogenic variant. Recurrence of disease due to either gonadal or parental mosaicism has been described in six previous individuals from three unrelated families [2,11,12];-In addition to the abovementioned distinctive features, MS is invariably associated with developmental delay and intellectual disability (DD/ID) [2,13,14,15], ranging from moderate to severe, even though mild ID has been rarely reported. Behavioral abnormalities, such as anxiety, autistic traits, hyperactivity, hetero- and auto- aggressivity and noise sensitivity are also reported in previous studies [2,3,13,14]. The sensory difficulties may worsen the severity of behavioral and psychiatric symptoms, mainly auditory hypersensitivity, that may increase anxiety, trigger challenging behavior [2], impair sustained attention and disrupt task performance [16].

A cross-sectional study comparing seven MS individuals to eight MSS individuals was performed to add information about cognition, adaptive behavior and sensory processing of both the conditions [17]. The study revealed ID with an overall low level of adaptive functioning in all MS participants. Sensory processing difficulties were always present and associated with problems in coping with stress and novelty. Nonetheless, a significant variability between the conditions and even among individuals affected with the same disease has been noticed. On the whole, cognitive, sensory and behavioral deficits are common in MS, hampering development, daily activities and social participation and often causing a severe adaptive functioning impairment. Limitations in adaptive behavior may contribute to the severity of the disability associated with MS, as established by the Diagnostic and Statistical Manual of Mental Disorders (DSM-5) [18], requiring the concomitant evaluation of adaptive behavior to establish the severity of ID.

Even though it is well acknowledged that cognitive and neuropsychological phenotype assessment is essential for timely diagnosis and early intervention, an accurate profiling of subjects with MS is still lacking. Except for the study performed by Mulder and colleagues [17], there are no other MS cohorts that have been systematically investigated for a neuropsychological profiling with standardized methods, which is a prerequisite for comparability over time, especially in the case of rare disorders.

Our study aims to further define the cognitive and adaptive behavior profile of 15 MS individuals, their linguistic skills and visuomotor integration abilities, providing quantitative data from standardized evaluations.

Specifically, our primary goal is to characterize an MS neuropsychological phenotype. Our study also widens the range of cases already described in the literature [2,17], with nine new MS individuals.

Both cognitive and adaptive behavior profiles were impaired in all MS subjects. We also predict poor performance in language and visuomotor integration tests, contributing to MS global impairment.

## 2. Materials and Methods

### 2.1. Participants

Fifteen Italian subjects with molecularly confirmed diagnosis of MS (M/F = 9/6), whose pediatric management has already been described [19], were included in this study. The clinical phenotype of 6 individuals of this cohort (Subjects 3, 4, 6, 8, 11 and 12, Table 1) was reported by Priolo et al. [2], while 9 individuals (Subjects 1, 2, 5, 7, 9, 10, 13, 14, 15, Table 1) had not previously been described. In all cases, an experienced medical geneticist and/or pediatrician performed the clinical suspicion of MS. All the patients were screened for mutations in the entire coding sequence of NFIX. The study is based on a retrospective data archive. Data refer to the database created for medical practice. The data relate to the Child and Adolescent Psychiatry Unit of the Bambino Gesù Children’s Hospital (Rome, Italy) and refer to a period of time between October 2020 and January 2022. Specifically, our study is a retrospective research in which we analyzed demographic, neuropsychological, psychological and linguistic information, extracting data from a database in which we habitually enter the score of all patients, coding each child with a serial number in order to guarantee privacy.

All parents gave informed e-consent to use data for research purpose. The demographic and clinical characteristics of the group are described in Table 1. Main MS features and their frequencies for the present cohort are reported in Appendix A. Genotypes of the patients reported are detailed in Appendix A.

### 2.2. Neuropsychological Assessment

Clinical data refer to evaluations performed from October 2020 to January 2022. Tests were administered during routine clinical activities, usually lasting 3 working days. The assessment was conducted by a team of trained and specialized child psychiatrists, psychologists and speech and language therapist and consisted of clinical observations, standardized evaluations and parent interviews. All individuals underwent a detailed evaluation aimed to define the cognitive profile, adaptive functioning, language skills and visuomotor integration.

#### 2.2.1. Cognitive Assessment

According to clinical practice, cognitive profiles were preferentially assessed through the cognitive battery of Leiter International Performance Scale-Third Edition (Leiter-3) [20], which is administered to subjects with known language impairment. Two individuals were evaluated using different standardized cognitive tests: subject 11, a verbally fluent 15.5 year-old girl, was assessed through Wechsler Intelligence Scale for Children–Fourth Edition (WISC-IV) [21], while subject 1, a 2.7 year-old female toddler, was assessed through Griffith Mental Development Scales—Third Edition (GMDS-3) [22] according to age (Table 1).

Leiter-3 is a tool for assessing nonverbal cognitive (cognitive battery), memory and attention abilities (memory/attention battery), designed to be administered to individuals without language skills from 3 to 75+ years. Cognitive battery provides a non-verbal intelligence quotient (NVIQ; herein after IQ).

WISC-IV allows assessment of the cognitive level of individuals ranging from 6 years to 16 years and 11 months. It provides a Full-Scale Intelligence Quotient (IQ) corresponding to the overall level of intelligence, together with four main reasoning indices (Verbal Comprehension VCI, Perceptual Reasoning PRI, Working Memory WMI, and Processing Speed PSI). In our study, PRI has been used to compare the individual assessed through WISC-IV to the remaining subjects’ IQs.

GDMS-3 is intended for children from birth to 72 months. It provides an overall score of developmental level (developmental quotient, DQ), as well as development quotients across five subscales, namely (A) Foundations of Learning, (B) Language and Communication, (C) Eye and Hand Coordination, (D) Personal–Social–Emotional and (E) Gross Motor domains. We used DQ to compare the youngest child to the other patients in our cohort.

Finally, we calculated mental age (MA) for each patient. The conversion from IQ to MA was helpful for a better understanding of language skills assessment results (see “Language assessment” session).

#### 2.2.2. Adaptive Behavior Assessment

Adaptive functioning was assessed using Vineland Adaptive Behavior Scales–Second Edition (VABS-II), a standardized semi-structured interview developed to support the diagnosis of ID according to DSM-5 for the assessment of the adaptive behavior in individuals from birth to 90 years and 11 months [23]. VABS-II is a caregiver interview and yields three domain scores: Communication, Socialization and Daily Living Skills (the fourth, Motor Skills domain, is investigated only in children younger than 7 years of age). Each domain is composed of specific subdomains: Communication (Expressive; Receptive; Written), Daily Living Skills (Personal; Domestic; Community), Socialization (Interpersonal; Play and Leisure; Coping Skills), Motor Skills (Fine Motor; Gross Motor). VABS-II also provides an overall Adaptive Behavior Composite (ABC) score, which is calculated by summing up the three (or four, in the case of the Motor Skills domain evaluation according to age) domains scores. VABS-II-ABC and relative domains provide age-based standard scores (mean, M = 100, SD = 15).

#### 2.2.3. Language Assessment

Language skills assessment in MS individuals was a challenge as they present with both cognitive deficit and language delay, which can be severe, making the choice of the evaluation tool very hard. Unfortunately, the majority of standardized tests do not provide scaled scores that can also be administered to individuals with ID and/or severe language delays. Language tests usually do not include a larger number of items to provide valid standardized scores for the lowest ranges. To overcome this psychometric issue, we decided to administer different language scales specifically selected for each subject, according to each equivalent MA, as provided by the above-mentioned intelligence tests. According to clinical practice, the following tests were used: (1) the Picture Naming Game (PiNG) test, which has been administered to the most verbally impaired children to assess lexical comprehension and production in Italian toddlers aged from 19 to 37 months [24]. PiNG test is divided into four blocks: Noun Comprehension, Noun Production, Predicate Comprehension and Predicate Production. For the purpose of our study, we decided to administer only the first two blocks. (2) The Phono-Vocabulary Test (Test Fonolessicale—TFL), which evaluates receptive and expressive vocabulary in children from 2 years and 5 months of age up to 6 years of age [25]. The test consists of 45 tables with four images each: a target, a phonological distractor, a semantic distractor and a non-related distractor. (3) The Battery for Assessing Language in Children Aged 4 to 12 (BVL_4-12, Batteria per la Valutazione del Linguaggio in Bambini dai 4 ai 12 Anni), which was used to evaluate articulatory and phonological discrimination skills. Specifically, the battery evaluates phonological, lexical, semantic, pragmatic and discursive skills in production, comprehension and repetition tasks [26]. We decided to administer the lexical and morphosyntactic comprehension and lexical production tasks to our cohort. (4) The Test for Reception of Grammar–Second edition (TROG-2), which is a picture selection test consisting of 80 sentences that must be coupled with the related pictures by choosing one picture out of four. It can be administered to people from 4 years of age and gives specific information about morphosyntactic comprehension [27].

#### 2.2.4. Visuomotor Integration Assessment

Visuomotor integration was assessed through the Developmental Test of Visual-Motor Integration–Sixth Edition (VMI). VMI is a standardized, norm-referenced assessment involving copying geometric forms to determine the visuomotor integration in children and adolescents from 3 years up to 18 years of age. The test includes two main versions, a full and a short form, plus supplemental tests [28]. We administered the full form but not the additional items.

### 2.3. Statistical Analysis

Descriptive statistics (M, min–max, SD) were calculated for age, IQ and VABS-II ABC domain and subdomain scores. Raw scores of IQ and VABS-II domains were converted into standard scores (M of 100 and SD of 15), while VABS-II subdomain raw scores were converted into v-scale scores (M of 15 and SD of 3). Normalized scores were used in all analyses.

To explore strengths and weaknesses within adaptive functioning in MS, the Wilcoxon matched-pairs test was used to test differences between pairs of VABS II domains and subdomains. Bonferroni’s adjustment was applied to the Wilcoxon matched-pairs test to control multiple comparisons.

Finally, to test for rank order relationship between IQ and VABS-II domains and between VABS-II- Daily Living Skills subdomains and VMI scores, we determined two-tails correlations using Spearman rank-order correlations coefficient (r_s_). The magnitudes of the correlation coefficients were stratified into groups as follows: small (0.1 < r < 0.3), moderate (0.3 < r < 0.5), large (0.5 < r < 0.7), very large (0.7 < r < 0.9) and nearly perfect (r > 0.9). A *p* value ≤ 0.05 was considered as statistically significant. All data analyses were performed using STATISTICA Six Sigma, STATISTICA release 7 (StatSoft, Inc., Tulsa, OK, USA, 1984–2006).

## 3. Results

### 3.1. Descriptive Analysis

A total of 15 patients were analyzed, 9 males and 6 females, aged between 2.7 and 25.6 years, all with clinical and molecularly confirmed diagnoses of MS. No significant genotype–phenotype correlation was observed when comparing NFIX variants with cognitive and adaptive functioning scores.

MS subjects were characterized by a low mean IQ score measured with either Leiter-3 (13 individuals, M = 52.4, SD = 9.1) or WISC-IV (1 individual, QIT < 40, VCI 50, PRI 50, WMI 49, PSI 47). One individual was assessed with the GMDS-3 showing low DQ (70). Our cohort documented very low intellectual levels: specifically, 67% of subjects (10/15) presented with an IQ below 3 SDS compared to the control population, and 33% (5/15) had an IQ between −3 and −2 SDS below the mean. Results on adaptive behavior showed an overall low functioning (M = 39.0); 10 individuals out of 15 (67%) showed severely impaired adaptive behavior (ABC < 4 SDS below the mean), 3 individuals (20%) had moderately impaired adaptive behavior (ABC between −4 and −3 SDS below the mean) and 2 individuals had conserved adaptive functioning (13%, ABC > −1 SD from the mean).

According to the cognitive and adaptive functioning assessments, one individual (Subject 10 Table 1) received a clinical diagnosis of mild ID, five individuals (Subjects 2, 4, 8, 11 and 12) were diagnosed with moderate ID and eight individuals (Subjects 3, 5, 6, 7, 9, 13, 14 and 15) with severe ID. The patient assessed through GMDS-3 (Subject 1) received a diagnosis of psychomotor delay.

As already underlined in the section “Neuropsychological Assessment”, language assessment required heterogeneous tests, which were selected for each subject according to MA. Additionally, a normalization between MA and language development, subtracting MA from the language development age, was performed for each subject. This normalization was performed for each language domain (Lexical Comprehension Age, LCA, Δ_LCA-MA_; Lexical Production Age, LPA, Δ_LPA-MA_; Morphosyntactic Comprehension Age, MCA, Δ_MCA-MA_). As a result of this additional strategy, we evidenced a significant impairment in language skills, which was even lower than MA. Specifically, Δ_LCA-MA_ and Δ_LPA-MA_ were greater than 2 years for the 53% of individuals and were approximately 1 year to 1 year and 11 months in the 20% and 27% of the other individuals, respectively. A Δ_LCA-MA_ from 0 months to 11 months was observable in 27% of participants, while only 20% exhibited this gap in Δ_LPA-MA_. In addition, all the participants displayed Δ_MCA-MA_ of more than 2 years.

Finally, since morphosyntactic production was not evaluable for the majority of MS individuals, we also carried out a qualitative analysis: 7/15 subjects (47%) had a prevalent use of olophrases (a single word expressing a complex idea, i.e., “water” for “I want water”), 5/15 subjects (33%) showed telegraphic language (composed by two or three words) and had a phonologic disorder, while 3/15 subjects (20%) exhibited language characterized by relatively complex morphosyntactic structures, although in the presence of critical phono-articulatory difficulties (Table 1). These results suggest that language comprehension is relatively more preserved than production in MS individuals.

Table 1 depicts demographic features (age, gender), cognitive abilities (IQ, PRI and DQ), VABS II-ABC, Δ_LCA-MA_, Δ_LPA-MA_, Δ_MCA-MA_ and qualitative analysis of language production and visuomotor integration abilities.

### 3.2. Adaptive Domains and Subdomains in VABS II

Descriptive statistics (MED, median, M, min–max, SD) for adaptive behavior are depicted in Table 2 (VABS II domains) and in Table 3 (VABS II subdomains). Results showed an overall low functioning (M = 39.0), with a distribution characterized by a variability (SD = 25) almost twice that of the general population. The Communication domain is the most affected (M = 35.4; SD = 22.2), followed by the Daily Living Skills domain (M = 45.7; SD = 19.7) and, lastly, by the Socialization domain (M = 52.3; SD = 25.4).

As Figure 1 shows, no differences between Socialization and Daily Living Skills were found (*p* > 0.05). On the other hand, the Daily Life Skills score was significantly higher than the Communication one (*p* ≤ 0.05). A clinically significant difference between Communication and Socialization scores was revealed, with the first remaining dramatically lower even after Bonferroni’s adjustment (*p* ≤ 0.05).

Taking into account VABS-II intradomain comparisons (Figure 2), no statistically relevant differences emerged between Socialization–Coping Skills and Socialization–Interpersonal and between Socialization–Coping Skills and Socialization–Play and Leisure (*p* > 0.05, in all comparisons). Instead, Socialization–Interpersonal vs. Socialization–Play and Leisure had almost quite average results with higher scores on the last subdomain (*p* = 0.06).

Regarding Communication subdomains, the comparison between Receptive and Expressive eluded the conventional threshold of significance, with the first being largely greater (*p* ≤ 0.05). Similarly, Receptive subdomain scores were significantly higher than Written subdomain scores (*p* ≤ 0.05), while the comparison between Expressive and Written scores narrowly missed achieving significance (*p* = 0.06).

When evaluating the Daily Living Skills subdomains, the differences between Personal and Community bordered on being significant (*p* = 0.06). Finally, performances on the Domestic subdomain were worryingly statistically higher than those on the Community and Personal subdomains (*p* ≤ 0.05, in all comparisons).

### 3.3. Correlation between Cognitive Abilities and Adaptive Behavior

Spearman’s rank-order correlations analysis revealed significant associations between IQ and all VABS-II domains (*p* ≤ 0.05, in all comparisons), indicating that if IQ scores increase as adaptive skills, all the VABS-II domains get better. More specifically, the magnitude of the relationship was large for IQ-VABS-II Communication (r = 0.7) and very large in the cases of IQ-VABS-II Daily Living Skills (r = 0.8) and IQ-VABS-II Socialization (r = 0.8).

### 3.4. Correlation between VABS-II-Daily Living Skills Subdomains and VMI-Visuomotor Integration

In order to investigate a correlation between Daily Living Skills and visuomotor integration abilities, we performed Spearman’s rank-order correlations between VABS-II-Daily Living Skills Total and subdomains and VMI scores. Our analysis showed that the correlation between VMI scores and Daily Living Skills scores very closely brushed the limit of statistical significance (*p* = 0.06). Instead, the relationship between Domestic and VMI failed to meet the statistical threshold (*p* > 0.05), while a significantly very strong correlation was found between VMI and both the Community and Personal domains (r > 0.7 and *p* ≤ 0.05).

## 4. Discussion

MS is an ultra-rare condition for which a detailed overview of the neuropsychological aspects is still lacking.

In the 80 cases of MS described so far [2,17], ID is always observed, ranging from mild to severe. However, to the best of our knowledge, there are no studies of standardized neuropsychological assessments in MS, with the exception of the study by Mulder and colleagues [17]. Our retrospective study aimed to further contribute to the neuropsychological characterization of MS, providing a quantitative description of the cognitive profiles, adaptive functioning, and linguistic and visuomotor integration skills of 15 affected individuals, which constitutes a novelty in the panorama of knowledge on MS.

When we assessed the cognitive level, the primary choice of a non-verbal cognitive test (Leiter-3) was guided either by our clinical experience and the literature reports, being aware that language impairment strongly interferes with the administration of scales that require the linguistic involvement skills. In our view, the possibility of smoothening the strength of the impaired language skills to better assess the cognitive level is the strongest point of this study. As predicted, our analysis showed that all the individuals presented with ID ranging from mild to moderate, with the prevalence of the latter being highest (10/15, 67%), in accordance with previous reports in the literature [2,15]. In addition, since the DSM-5 definition of ID [18] includes not only below average IQ, but also adaptive functioning deficits, we investigated both aspects using standardized tests (VABS-II). As predicted, 13 out of 15 subjects showed an adaptive functioning lower than normal, ranging from moderately to severely impaired, in line with their IQs.

Indeed, not surprisingly, analysis of IQ-VABS-II domain correlations, showed proportional increase in IQ with respect to adaptive behavior across all domains (ABC, Communication, Daily Life Skills and Socialization).

Regarding IQ-VABS-II domain relationships, an interesting point is that VABS-II ABC scores were lower than IQ for all individuals, with the exception of two subjects who showed an in-average profile (subjects 1 and 10), likely due to several factors impacting on adaptive functioning, from environmental factors (e.g., presence or lack of family and social support, educational and rehabilitative interventions) to any other physical or neurosensory deficit.

Considering VABS-II domain and subdomain comparisons, as in the work of Mulder and colleagues [17], the Communication domain was the most affected. The poor performances on the Communication Scale are not surprising as they are in line with the striking language difficulties in MS individuals. On the other hand, the observation of less impairment in social skills could argue with the previously reported prevalence of autistic traits in patients with MS [17]. We noticed a preserved communicative intentionality despite the linguistic impairment, which could mean that MS social difficulties may be mostly due to the severe language deficits rather than to the lack of interest in interpersonal relationships typical of autism. As further proof, performances on the Receptive subdomain inside the Communication domain were significantly higher than the ones on the Expressive subdomain, which could indicate that MS individuals keep some ability to understand and may want to communicate and socialize with others, but they do not have the expressive skills required to be in relationships. This alternative interpretation broadens possibilities regarding treatment options. For instance, a speech therapy rehabilitation that includes AAC (augmentative and alternative communication) could accommodate the communicative desire of MS patients hindered by language difficulties.

Concerning the Socialization domain alone, we found out that there were no differences between the subdomains and that it was the least impaired among VABS domains, even though v-scale scores on all Socialization subdomains were always more than 2 SD below the mean. In deep analysis, a borderline level of statistical significance was observed only in the comparison between Interpersonal and Play and Leisure, with better performances in the latter. A possible explanation of this finding is that language deficiency impacts all the other Socialization subscales that require communication skills. Regarding Daily Living Skills domain, scores were dramatically lower in the Personal and Community subscales with respect to the Domestic subdomain. The first two subdomains may be more demanding in terms of the complexity of motor schemes, actions and praxis skills necessary to perform these items. Our results are partially in contrast with what was reported by Mulder and colleagues [17], who found that Daily Living Skills were relatively well developed and that Social Adaptive Behavior was dramatically low. This divergence may be due to the relatively different mean age in our cohort (11.9 ± 5.8 years) with respect to that of Mulder’s (14.6 ± 6.7 years). It is possible that, while Daily Living Skills gradually improve with the time, the learning curve for socialization levels out and reaches its peak. An alternative explanation could be that the two study cohorts exhibited different psychopathological characteristics. Indeed, we know that both anxiety and autistic traits can be present in MS, with a high individual variability. Anxiety, and in particular social anxiety, the lack of interest in interpersonal relationships and the atypical communication may resemble autism and may impair functioning in the Socialization domain. Mulder and colleagues [17] and the present study failed to psychopathologically characterize MS individuals, but its role in limiting adaptive functioning in the Socialization domain can be hypothesized. Further studies and follow-up assessments are required to verify this hypothesis.

In addition to the assessment of cognitive and adaptive profiles, we also evaluated visuomotor integration abilities. When considering VMI scores alone, we observed dramatically poor MS performances. This result is not surprising since visuomotor integration allows coordination of visual data about the body, the extracorporeal space and the relative position of the body in space, with movements of the whole body or part of it, in order to guarantee coordinated, smooth and effective motor sequences. MS subjects displayed impaired visuomotor, visuoperceptive and visuospatial integration, which may be attributable to ocular motility abnormalities and lower visual acuity and/or to cortical-subcortical visual integration deficits [19], in line with some other previous reports [7].

Another intriguing aspect of the data is the statistically significant correlation between VMI scores and the Personal and Community subdomains scores but not between VMI scores and the Domestic subdomain scores. The explanation may be dual: on one side, the items listed in the Domestic subdomain may be coarser because of the required praxis skills, and therefore they do not correlate with the visuomotor integration skills which, in contrast, play a greater role in finer items such as those included in the Personal and Community subdomains; on the other side, we cannot exclude that parents are less demanding at home, not allowing those symptoms to be as evident as if they were in other environments. Nevertheless, these different possible explanations are not mutually exclusive.

The greater gap between the receptive profile of the Communication domain and the expressive one is not detected by the structured language assessment tests either; these latter ones, however, unlike the VABS II Communication Domain, analyze the micro-elaborative level (lexical and morphosyntactic abilities) of the language rather than the macro-elaborative one (pragmatic, narrative, and dialogic skills) and, in a broader sense, the communicative functionality. Indeed, structured language evaluation has been a challenging task as MS individuals exhibit a large variability in cognitive profile and language development. In order to collect reliable information on language skills for each subject, we selected different language scales according to each individual MA. This strategy allowed us to better characterize the micro-elaborative levels of language within each subject, even though it invariably impacted comparability and interpretation of results among single individuals by using different measures, so we cannot exclude that the adoption of heterogeneous scales impacted our results. Additionally, we observed that receptive language development was always lower than MA, even though more preserved than the expressive one, thus confirming Mulder’s results. Furthermore, speech was so damaged that morphosyntactic production was not evaluable for most MS children, forcing us to perform only qualitative analysis.

Our study has strengths and limitations due to the peculiar clinical features of this ultra-rare condition. The sample size could be considered small from a purely statistical point of view but representative when considering MS prevalence. Furthermore, we did not use homogenous assessment tests, which inevitably prevented direct comparisons between subjects. On the other side, choosing tests according to individual age, cognitive profile, linguistic skills, and compliance allowed us not to resort to any adaptation or facilitation in administering the tests themselves, which is different from previous reports [17]. Another limitation of our work is the lack of a control group: our analyses focused on comparisons with reference data. Future studies should be designed to include a control group and enlarge the sample size to obtain more consistent data and to characterize the cognitive profile in MS.

Our findings underline the importance of a detailed neuropsychiatric evaluation in individuals affected with MS. A systematic assessment of cognitive and adaptive abilities is essential to better characterize the neuropsychological phenotype of this ultra-rare disorder. A disorder characterization exploring both cognitive and adaptive profiles in larger cohorts will potentially represent a diagnostic aid for MS individuals. Lastly, longitudinal studies should be carried out in order to have a clearer description of MS profiles and to investigate adulthood evolution in terms of possible regression of cognitive, language, and adaptive functioning.

## 5. Conclusions

Individuals with MS are often referred to neuropsychiatric counseling because of DD involving motor and language skills. Characteristic phenotypic features and macrocephaly can clinically address the diagnosis, but a clinical genetic evaluation and subsequent molecular confirmation are mandatory to diagnose MS. A multidisciplinary approach is essential to cope with the disability itself and to acquire and develop skills to ensure the best possible functioning in daily life.

As disability in MS may be worsened by concurrent psychologic and psychiatric comorbidities (e.g., anxiety, autistic traits), further research should aim to analyze the behavioral and psychopathological profiles of MS and its influence on cognitive outcomes, adaptive functioning, and communication skills in order to promptly prevent or treat psychiatric comorbidities, thereby improving the outcome for affected individuals.

## Figures and Tables

**Figure 1 jcm-11-04078-f001:**
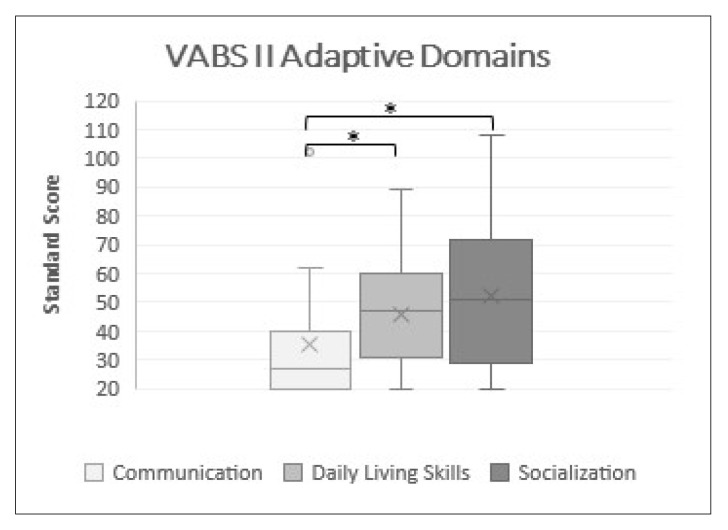
Domain comparisons. Box plots of standard score of VABS II domains, * significant at *p* ≤ 0.05.

**Figure 2 jcm-11-04078-f002:**
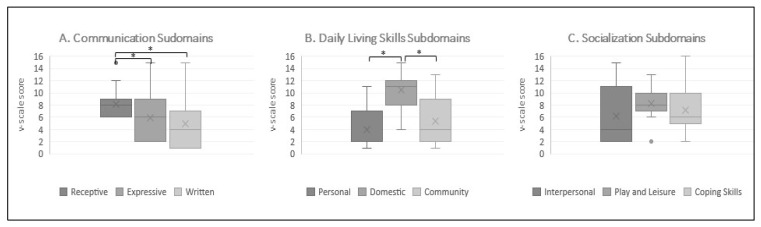
Intradomain comparisons (**A**) Box plots of v-scale scores of Communication subdomains (**B**) Box plots of v-scale scores of Daily Living Skills subdomains (**C**) Box plots of v-scale scores of Socialization subdomains, * significant at *p* ≤ 0.05.

**Table 1 jcm-11-04078-t001:** Demographic features, cognitive abilities, adaptive behaviors, language profiles and visuomotor integration abilities of studied cohort.

N	Gender	Age	IQ	ABC	$\Delta_{LCA-MA}$	$\Delta_{LPA-MA}$	$\Delta_{MCA-MA}$	Language Production: Qualitative Analysis	VMI
1	F	2.7	70 *	87	0	n.a.	n.a.	prevalent use of olophrases	n.a.
2	M	6.4	65	39	−2.1	−2.2	n.a.	prevalent use of olophrases	45
3	F	7.3	54	23	−0.8	n.a.	n.a.	prevalent use of olophrases	45
4	M	7.11	58	48	−1.3	−1.2	n.a.	use of telegraphic language, phonological disorder	45
5	M	8.1	52	42	−1.2	−1.2	n.a.	use of telegraphic language, phonological disorder	45
6	F	8.7	54	31	−1.6	n.a.	n.a	prevalent use of olophrases	45
7	M	8.11	40	33	−0.1	−1	n.a.	prevalent use of olophrases	n.a.
8	M	11.3	62	56	−0.8	0	−2.8	use of complex sentence, phonological disorders	72
9	M	13.3	42	20	−2.5	n.a.	n.a	prevalent use of olophrases	45
10	M	13.10	69	100	−3.8	−3.7	−3.9	use of complex sentences, phonological disorders	65
11	F	15.5	50 **	21	−2.7	−0.8	−3.4	use of complex sentences, phonological disorders	49
12	M	16.2	48	25	−2.6	−0.3	−3.6	use of telegraphic language, phonological disorder	59
13	M	16.4	45	20	−2.7	n.a.	−3.9	prevalent use of olophrases	45
14	F	17.6	47	20	−3.1	−1.3	−4.2	use of telegraphic language, phonological disorder	47
15	F	25.6	40	20	−7.3	−7.5	−7.4	use of telegraphic language, phonological disorder	45

Legend: M, male; F, female; IQ, intelligence quotient, * developmental quotient, ** perceptual reasoning index; VMI visuomotor integration; n.a., not available due to difficulties subjects showed.

**Table 2 jcm-11-04078-t002:** Descriptive statistic in VABS II Domains.

VABS II Domains	MED	M	Min–Max	SD
Communication	27.0	35.4	20.0–102.0	22.2
Daily Living Skills	47.0	45.7	20.0–89.0	19.7
Socialization	51.0	52.3	20.0–108.0	25.4
Motor Skills	54.5	54.5	33.0–76.0	30.4
ABC	31.0	39.0	20.0–100.0	25.0

Legend: MED, median; M, media; Min–Max, minimum–maximum; SD, standard deviation.

**Table 3 jcm-11-04078-t003:** Descriptive statistic in VABS II Subdomains.

VABS II Subdomains	MED	M	Min–Max	SD
Receptive	8.0	8.1	6.0–15.0	2.5
Expressive	6.0	5.9	6.0–15.0	4.0
Written	4.0	4.3	1.0–14.0	7.1
Personal	2.0	3.9	1.0–11.0	3.0
Domestic	11.0	10.5	4.0–15.0	2.8
Community	4.0	5.4	1.0–13.0	3.8
Interpersonal	4.0	6.3	2.0–15.0	4.4
Play and Leisure	8.0	8.3	2.0–17.0	3.7
Coping Skills	6.0	7.2	2.0–16.0	3.9

Legend: MED, median; M, media; Min–Max, minimum–maximum; SD, standard deviation.

## Data Availability

The data that support the findings of this study are available on request from the corresponding author. The data are not publicly available due to privacy or ethical restrictions.

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
