# Peer review of "Characterization of Cognitive, Language and Adaptive Profiles of Children and Adolescents with Malan Syndrome"

_jcm, 2022, doi:10.3390/jcm11144078_

Round 1

Reviewer 1 Report

Alfieri and colleagues nicely describe the cognitive, language, and adaptive profiles of individuals with Malan syndrome (MS). This is valuable for individuals and families impacted by MS, as well as clinicians and other health providers working with MS.

While all the information provided is interesting and presented correctly, it feels like a lot of patient information is missing. Its possible that this information is in the submitted manuscript that is cited several times. I think a table/supplemental table with all the clinical features of each patient is necessary. Additionally, the genetic NFIX variants should be included if possible. It would be interesting to see if there are any differences in the behavioral/cognitive phenotypes by variant. 

VABS II scores would be useful to compare to the normal population (statistics would be great, but even just a line showing the normative values would help). Comparing to the Mulder study would be nice as well. Since they are allelic, if you can compare to MSS that would be interesting. 

Just a few small things: 

the gene name is not always in italics

Table 1 is referenced in multiple type faces (italics, lower case, etc). Just needs to be consistent

The Tables may have some mis-formating (the first column looked jumbled)

It would be helpful to define olophrases (olophrasis? Both are used) like you did for telegraphic language

You don’t need “panel” in Figure 2. Just A B C is fine. Listing what statistical test was used for each figure would be good.

Overall this is a useful paper for those involved with MS and related disorders. With a better understanding of the cognitive profile, better therapeutic strategies may be developed. 

Reviewer 2 Report

In this study, the authors performed a standardized neuropsychological assessment of cognitive, language, and adaptive profiles in a relatively large sample of young patients diagnosed with Malan Syndrome.

This characterization represents a useful resource to clinicians that could facilitate the identification of behavioral issues and appropriate treatment options.

The research objectives are well defined and the design of the study is appropriate. The methods and results are properly described. The introduction and discussion sections are informative. The conclusions are in general supported by the data presented.

My specific comments are reported below.

1.     “MS is allelic to Marshall-Smith syndrome”. I am not sure what “allelic to” means in this sentence.

2.     In figures 1 and 2 please clarify what the error bars represent.

3.     Since the article by Macchiaiolo et al. describing the pediatric management of these patients is currently not available, it would be great to provide at least a brief description of the type of management and treatments these patients underwent to. This would also provide more context to the neurobehavioral data presented in the results.
